# Reverse translation of adverse event reports paves the way for de-risking preclinical off-targets

Mateusz Maciejewski[1]*[†], Eugen Lounkine[1], Steven Whitebread[1], Pierre Farmer[2], William DuMouchel[3], Brian K Shoichet[4]*, Laszlo Urban[1]*

[1]Novartis Institutes for Biomedical Research, Cambridge, United States; [2]Novartis Institutes for Biomedical Research, Basel, Switzerland; [3]Oracle Health Sciences, Oracle Health Sciences, Burlington, United States; [4]University of California, San Francisco, United States

**Abstract** The Food and Drug Administration Adverse Event Reporting System (FAERS) remains the primary source for post-marketing pharmacovigilance. The system is largely un-curated, unstandardized, and lacks a method for linking drugs to the chemical structures of their active ingredients, increasing noise and artefactual trends. To address these problems, we mapped drugs to their ingredients and used natural language processing to classify and correlate drug events. Our analysis exposed key idiosyncrasies in FAERS, for example reports of thalidomide causing a deadly ADR when used against myeloma, a likely result of the disease itself; multiplications of the same report, unjustifiably increasing its importance; correlation of reported ADRs with public events, regulatory announcements, and with publications. Comparing the pharmacological, pharmacokinetic, and clinical ADR profiles of methylphenidate, aripiprazole, and risperidone, and of kinase drugs targeting the VEGF receptor, demonstrates how underlying molecular mechanisms can emerge from ADR co-analysis. The precautions and methods we describe may enable investigators to avoid confounding chemistry-based associations and reporting biases in FAERS, and illustrate how comparative analysis of ADRs can reveal underlying mechanisms.

*For correspondence: matt@ mattmaciejewski.com (MM); bshoichet@gmail.com (BKS); laszlo.urban@novartis.com (LU)

Present address: [†]Pfizer, Inc., Cambridge, United States

## Introduction

Safety assessment of drug candidates is crucial for drug discovery, enabling the development of medicines that achieve the desired therapeutic effects with the least risk of adverse side effects. Preclinical regulatory investigations and clinical trials are designed to address safety of drug candidates and eliminate those that do not meet risk-benefit expectations (*Cook et al., 2014*). However, limited access to large, diverse patient populations in clinical trials, untested drug co-administrations, as often occurs, especially with elderly patients on multiple medications, and development of ADRs associated with chronic treatment, often results in post-marketing labeling and occasional withdrawals (*Wysowski and Swartz, 2005*; *Lasser et al., 2002*; *Friedman et al., 1999*; *Downing et al., 2017*). Thus, postmarketing pharmacovigilance is essential to track ADRs and ultimately reduce over 1 million serious drug-related side effects that occur each year in the USA. Between 5% and 10% of these ADRs are fatal (*Lazarou et al., 1998*), and many others cause patient suffering, hospitalization, and increased health system burden (*Moore et al., 1998*). Indeed, the fatality rate attributed to ADRs puts them among the top causes of death in the USA (over 40,000 in 2011), similar to suicide-related mortality (*Hoyert and Xu, 2012*).

Determinant tools in post-marketing pharmacovigilance are databases that aggregate ADR reports. Foremost among these is the FDA Adverse Event Reporting System (FAERS), which is

**eLife digest** New treatments are tested in clinical trials before they are licensed for use in patients, but until the drugs are available for prescribing it's not always possible to identify every side effect. When the drugs enter the clinic, they might be prescribed to patients with multiple medical conditions, or combined with other treatments. The drugs may also be taken for longer periods of time than tested in trials. It is therefore common for new adverse reactions to emerge after a drug is in widespread use.

The FDA Adverse Event Reporting System (FAERS) is a surveillance system used in the United States for reporting drug side effects after new treatments have been licensed. Healthcare professionals and patients can submit reports to the database, logging the adverse drug reactions that they have experienced.

FAERS currently contains over 8.5 million entries, and is growing all the time. However, Maciejewski et al. show that the database has several shortcomings that are reducing its usefulness. For instance, on average any given drug will have 16 different names in the system; this makes it challenging to group all of the reported side effects so that trends and patterns can be correctly seen.

To address this first problem, Maciejewski et al. grouped together drugs according to their active ingredients, rather than their name. This made it much easier to account for subsequent, and more crucial conflating factors such as multiple reports for the same adverse event and patient, or cases where adverse reactions were confused with the diseases that the drugs are trying to treat. For example, diabetes was listed as a side effect for drugs used to treat diabetes.

Building on this cleaned-up dataset, Maciejewski et al. monitored how adverse event signals evolve over time and uncovered biases that were hard to see otherwise. For example, side-effects were reported more often when drugs were in the news. More strikingly, this bias affected not only the drug in question, but also other drugs that acted in the same way or on the same molecular target.

The computational method developed by Maciejewski et al. allows the data in FAERS to be combined and corrected, making easier to evaluate the safety of different medicines. The link between adverse side effects and the molecular targets of the drug, via the ingredient's chemical structure, furthermore makes it possible to analyze such clinical data reliably by using chemical and genetic information. In the future, this method could also help to identify previously unknown side effects and the biological mechanisms behind them. This could help researchers to develop new drugs with improved side effect profiles.

perhaps the most extensive, and among the most widely accessible of these databases, currently containing over 8.5 million reports and rapidly growing (*U.S. Food and Drug Administration, 2016*). FAERS and related databases, such as those of the EMEA and of Health Canada, can provide specific ADR phenotypes typical for either individual drug classes or specific indications and can be accessed either directly (*U.S. Food and Drug Administration, 2016*) or by APIs (*RELX Intellectual Properties SA, 2016*; *U.S. Food and Drug Administration, 2016*). These large-scale adverse event databases enable analysis to relate clinical phenotypes and compounds (*Tatonetti et al., 2012*), and they have been widely used by the clinical community with much impact (*O'Connell et al., 2006*; *Elashoff et al., 2011*; *Lawrence et al., 2006*; *Mackey et al., 2007*).

It is an attractive proposition to exploit the sheer scale of FAERS to detect drug-ADR associations that would otherwise be missed. A challenge in doing so has been the heterogeneous data sources and data conflation in the database.

FAERS, while providing a solid frame for reporting, contains redundancies, biases, and conflations that affect its analysis and interpretation (*McAdams et al., 2008*). Our ability to even correlate drugs with their effects is obscured by something as simple as the tangle of drug synonyms in FAERS - on average 16 different names for medicines containing each active drug ingredient - which can obscure associations.

Here, we investigate the effects that these data conflations, inflations, and inaccuracies can have on ADR and mechanistic inference from FAERS, and methods to address them. We begin by mapping drug identifiers in FAERS to normalized chemical structures of their ingredients, which brings together observations over the 'full drug', not just particular drug names and synonyms, which remain incomplete. Mapped to unique chemical structures, we could compute time-resolved profiles of drug-ADR associations, which revealed intriguing comorbidities and similarities of ADRs between drugs, and of their time evolution. We then turned to the origins of and controls for reporting biases in FAERS, considering stimulated reporting and the several different, often non-medical communities that can contribute to FAERS. This was facilitated by a time-evolution analysis of ADR reports, and its correlation with contemporary news events. We illustrate how these biases can ramify with in-depth analysis of FAERS content on two COX-2 inhibitors, rofecoxib and celecoxib, and with two PPAR-$\gamma$ agonists, rosiglitazone and pioglitazone. As examples of how these analyses can link ADRs to specific targets, we consider the differential ADR profiles of drugs used for the treatment of attention deficit hyperactivity disorder (ADHD), and how their distinct ADRs may be explained partly by molecular targets - a logic that is often used - combined with pharmacokinetic exposure - which is often overlooked. Similarly, we investigate the differentiation of the hypertensive side effects of VEGF-Receptor (VEGF-R2) inhibitors based on their potency and pharmacokinetic (PK) profiles. The precautions and methods we describe, may enable investigators to use FAERS with increased confidence and avoid confounding chemistry-based associations and reporting biases. This study also illustrates how comparative analysis of ADRs can reveal underlying mechanisms and highlight the reverse translation value in the drug discovery process.

## Results

### Analysis of content: unexpected trends in FAERS reporting

The FAERS database holds over 8.5 million reports and is steadily growing (over 1,320,000 reports added in 2015; *Figure 1A*. We extracted 8,749,375 FAERS reports, mapped to 7,095,566 individual cases. Often a patient's condition is monitored over a span of multiple reports, which must be considered when investigating the incidence of a particular drug-ADR association (*U.S. Food and Drug Administration, 2016*).

Inflation of reports by multiplication can increase the apparent significance of a drug – adverse effect association, particularly when the total number of reports is low. To systematically identify the most similar cases, we compared all pairs of reports using demographic and prescription data. Almost 1% of the reports in FAERS (61,780 cases) represent multiple entry cases with identical drugs, identical ADRs, event dates, patient age and gender (*Supplementary file 1*). Intriguingly, only half of the reports in FAERS were submitted by healthcare professionals (*Figure 1B*). Over one-third of them (3.2 million) were initiated by the patients themselves and 9% were labeled 'non-specified'. Lawyers reported 3% of all FAERS cases (*Figure 1B*).

FAERS uses seven descriptors of report outcomes: 'Death', 'Life-Threatening', 'Disability', 'Congenital Anomaly', 'Required Intervention to Prevent Permanent Impairment/Damage', 'Hospitalization – Initial or Prolonged', and 'Other'. Among these, only 'Other' is used to report relatively benign outcomes. Unexpectedly, only around 40% of the outcomes were identified as 'benign', whereas almost 15% of reported cases result in death (*Figure 1C*). It is a feature of reporting in an open submission database like FAERS that this ratio does not reflect the true balance between fatal and relatively benign drug ADRs, but rather the ratio of the ADRs that are thought to merit reporting.

Among the 945,526 reports where death is the outcome of the ADR, 42,526 were linked to cardiac arrest and 50,155 to suicide. Top molecular ingredients of drugs that were primary suspects in death reports were rosiglitazone: 17,165 (indication type II diabetes), rofecoxib: 11,386 (primary indications: arthritis, pain; withdrawn from the clinic), reteplase: 11,386 (indication of acute myocardial infarction (MI)), and thalidomide: 17,104 (indication of myeloma multiplex; additionally, 26,429 cases of death have been attributed to lenalidomide, a derivative of thalidomide also prescribed for myeloma). For drugs like rofecoxib or rosiglitazone, which are prescribed for manageable and non-life threatening diseases, the inference that the ADR has led to death can be reasonably made. Similarly, a comparison of celecoxib (reported number of deaths: 4,066; Standardized Mortality Ratio [SMR]

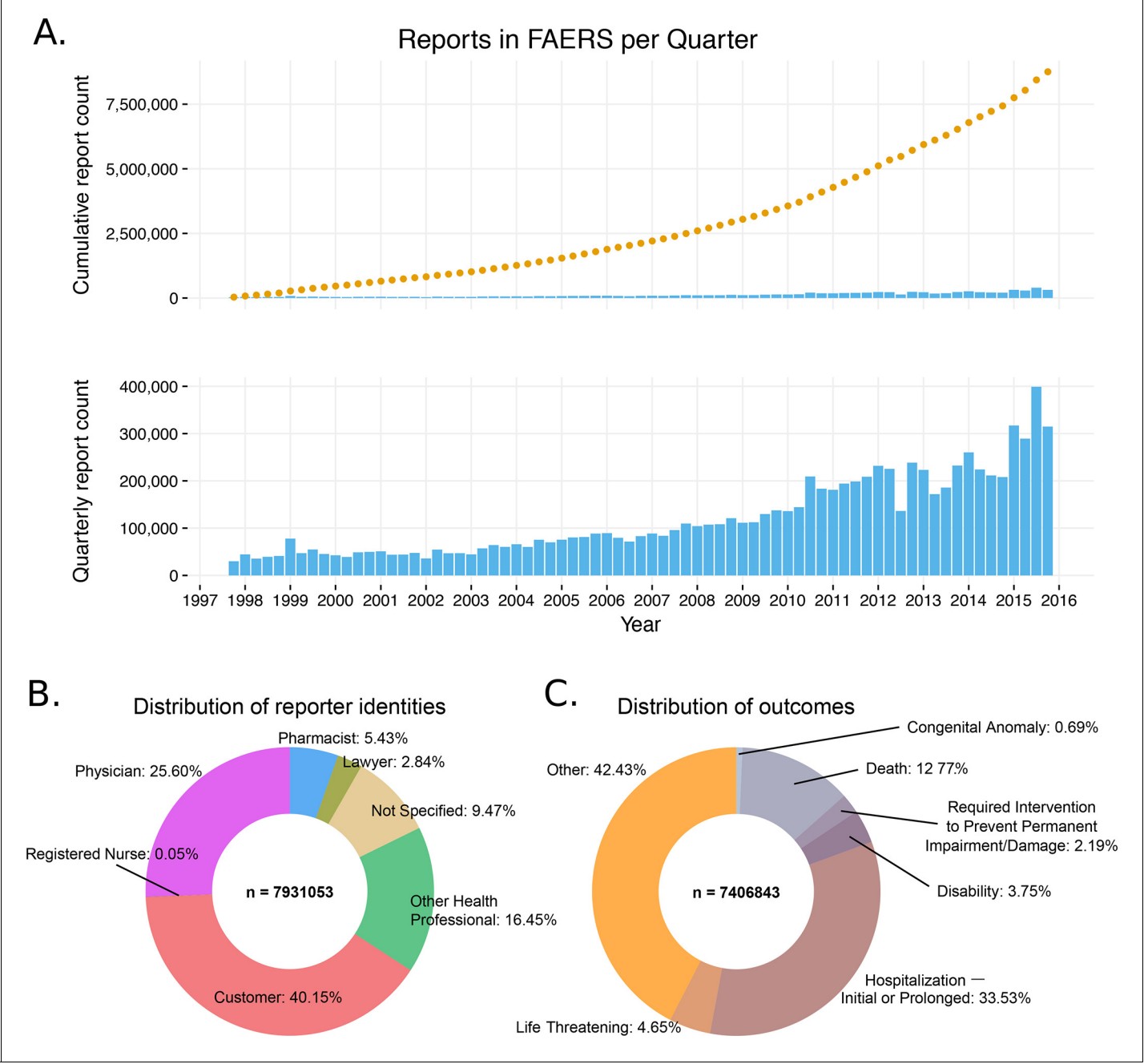

**Figure 1.** General information of the FDA Adverse Event Reporting System (FAERS) content (1997–2015). (**A**) The cumulative number of reports in FAERS is shown in the top panel; the bottom panel shows the number of new reports per quarter. (**B**) Distribution of reporter identities. Data are based on reports submitted between Q2 2002 (identification of reporting individuals started at this time) and Q4 2015. (**C**) Distribution of reports by the 7 ADR outcomes defined in FAERS.

[*Everitt and Skrondal, 2010*]: 1.3) and rofecoxib, which are prescribed for the same indication, highlights the significantly higher SMR of patients taking the latter drug (SMR: 5) (*Rostom et al., 2007*). However, the attribution of death as an ADR of thalidomide when it is used to treat myeloma multiplex, a life-threatening, malignant disease (*Singhal et al., 1999*) may be hard to support; it seems likely that the 'ADR' here reflects the cancer that the drug is meant to treat. Similarly, the acute myocardial infarction that reteplase is used to treat (*Wooster and Luzier, 1999*) may well be the cause

of many of the death ADRs with which the drug is tarred, not the drug itself. When a drug is used to treat a life-threatening disease, care is warranted in interpreting death as an ADR of that drug.

## Mapping drugs to their molecular ingredients improves signal retrieval

In most FAERS studies, drugs are identified using $R_x$Norm (*Wang et al., 2013*; *Nelson et al., 2011*), a set of drug synonyms supplied by the National Library of Medicine. This mapping is sufficient for the questions that may be asked of FAERS by a clinical professional, such as the safety signals for a particular drug formulation. However, products that have different identities in resources such as $R_x$Norm share common molecular ingredients and are highly similar in their activities on molecular targets. To investigate the ADRs associated with fluoxetine, for instance, one must aggregate its 378 different synonyms. Without such aggregation, well-known fluoxetine side effects such as sexual dysfunction become statistically insignificant (four cases when only the fluoxetine drug synonym Prozac is considered; Relative Reporting Ratio [RRR] = 1.75; q-value = 1), whereas once aggregated, these ADRs stand out clearly (87 cases; RRR = 6.67; q-value = $2.56 \cdot 10^{-96}$). Conversely, in its non-aggregated form, Prozac appears to have statistical significant associations with sex chromosome abnormality (one case; RRR = 2.96; q-value = $2 \cdot 10^{-3}$). Aggregated, however, this association becomes insignificant (one case; RRR = 2.78; q-value = 1). For those interested in the molecular basis of drug actions and side effects, a simple way to interrogate the drugs as molecules is critical.

Accordingly, we mapped the active drug ingredients in over 98% of the reports using a combination of natural language processing and multiple databases of synonyms (see Materials and methods). Not only does this value compare favorably to the 81% recognition achieved using only the synonyms alone in $R_x$Norm, but it allowed us to look for associations drawing on standard cheminformatics-based searches. Surprisingly, of the 2729 unique ingredients identified, only 1892 were annotated as a primary suspect in at least one report; said a different way, 837 active drug ingredients had no reported ADRs whatsoever. A plot of the ingredients that were associated with ADRs shows that an exponentially decaying distribution, with 90% of the ADRs attributed to 40% of the drug ingredients (*Figure 2*). After correction of distribution for ADRs with q-values better than 0.05, 90% may be attributed to 46% of the investigated drugs. This ingredient mapping was used throughout subsequent analyses (see Materials and methods and Supplementary Material).

As expected, mapping drugs to their active ingredients, and not simply relying on synonym aggregation, reinforced the strength of the drug-ADR signals. For example, the non-steroidal anti-inflammatory drug (NSAID) indomethacin is used to treat chronic pain and fever (*MedicinesComplete, 2014*). When we assessed indomethacin as an ingredient, a strong signal linked it with gastric ulcer (RRR = 10.40; q-value = $3.65 \cdot 10^{-72}$), and gastric ulcer hemorrhage (RRR = 7.99; q-value = $6.78 \cdot 10^{-18}$). These adverse events are known from the labels of indomethacin-containing drugs, also confirmed in World Drug Index (WDI) (*Thomson Scientific, 2016*). However, when we

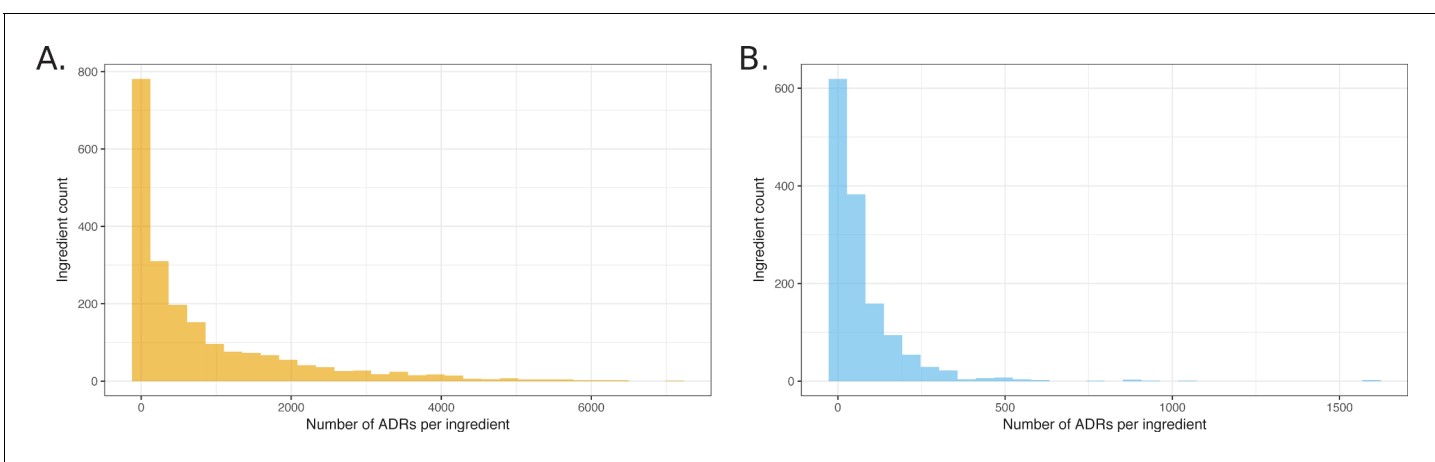

**Figure 2.** Histograms showing the distribution of the number of ADRs that were attributed to unique ingredients. (A) All observed ingredient – ADR pairs. (B) Pairs observed below the q-value cutoff of 0.05.

searched the trade names of the drugs in which indomethacin is used ($R_x$Norm synonym matching), these signals were dissipated in the noise: the strongest signal for gastric ulcer decreased to RRR = 1.79, q-value = 1.00; the strongest signal for gastric ulcer hemorrhage dropped to RRR = 2.42, q-value = 1.00.

## Bias in ADR reporting by indication, changes in regulatory, clinical, social and legal environment

Sometimes, ADRs are conflated with indications, and vice versa. An example is a report of rosiglitazone being prescribed for type two diabetes mellitus, with the ADR in the report being also diabetes mellitus (*Table 1*). In another report, rosiglitazone was identified as the primary suspect for congestive heart failure, as well as a therapeutic agent that was prescribed for the very same condition (*Table 1*). We quantified this indication bias both globally and over time. Approximately 5% of all reports for any drug describe the drug's indication as an adverse event. The number of reports in which the same ADR and indication was reported increased linearly with the increasing number of yearly reports until 2011, followed by a sudden drop (*Figure 3*). We could trace an FDA advisory 'refresher' presentation on guidelines of ADR reporting for clinical trials to an effective date of March 28, 2011 (*Devine, 2016*). This document provides clear instructions for submitters to distinguish between pre-existing conditions and ADRs and indeed may have had a significant effect on reporting quality.

We took a closer look at the reports of rosiglitazone, where occurrence of diabetes as a side effect was attributed to the usage of this drug relatively frequently until 2004 (this obviously erroneous association is significant if considered in the reporting window of rosiglitazone (until 2011), with an RRR = 1.57 and a q-value $<10^{-5}$). After 2004, this association decreased, as did the overall prescriptions and reporting of this drug, owing to its widely-reported cardiovascular side effects (*Mannucci et al., 2010*; *Nissen and Wolski, 2007*). In general, a simple comparison of indications and reported ADRs reduces the bias of verbatim repetition.

We applied these methods to investigate how reports for individual drugs change over time. In particular, we monitored the total number of reports filed and the incidence of adverse events preferentially reported at different time points. When reports are sorted by event dates in FAERS, 'spikes' occur on the first day of each month, and even larger spikes on the first day of each year. Importantly, drug-serious ADR signals show a time-dependent increase (see *Figures 4A*, *5A*, *6A*, and *Figure 7A*). The changes in drug-ADR associations over time can, of course, reflect new populations to which the drug is exposed.

We assessed the time evolution of reports of rofecoxib, a nonsteroidal anti-inflammatory drug (NSAID) that relieves pain through COX-2 inhibition (*Figure 4*). Several important events occurred over the clinical life of rofecoxib since its approval by the FDA in 1999: (1) A clinical study by Bombardier et al. published in November 2000 concluded that rofecoxib increased the risk of cardiovascular events (*Bombardier et al., 2000*). (2) Introduction of warnings for cardiovascular events on the labels of Vioxx (a brand name of rofecoxib) in April 2002. (3) Withdrawal of rofecoxib from the market on September 30th 2004.

**Table 1.** Confusion of ADRs with indications. Report and case numbers identify two FAERS reports where the ADR is confused with the indication.

For the first case, rosiglitazone prescribed for diabetes (Indication) is identified as the primary suspect (PS) for causing diabetes mellitus as an ADR as well. In the second case, 'cardiac failure congestive' is given as the indication for rosiglitazone with the reported ADR of 'cardiac failure congestive'. The third case exemplifies correct reporting, where both the ADR and the indication of rosiglitazone are reported correctly.

| Report | Case | ADR | Drug / Role / Indication |
|---|---|---|---|
| 6545021 | 179039 | Diabetes mellitus | Rosiglitazone/PS/Diabetes mellitus |
| 5521616 | 162007 | Cardiac failure congestive | Rosiglitazone/PS/Cardiac failure congestive |
| 6380841 | 7085373 | Cardiac failure congestive | Rosiglitazone/PS/Diabetes mellitus |

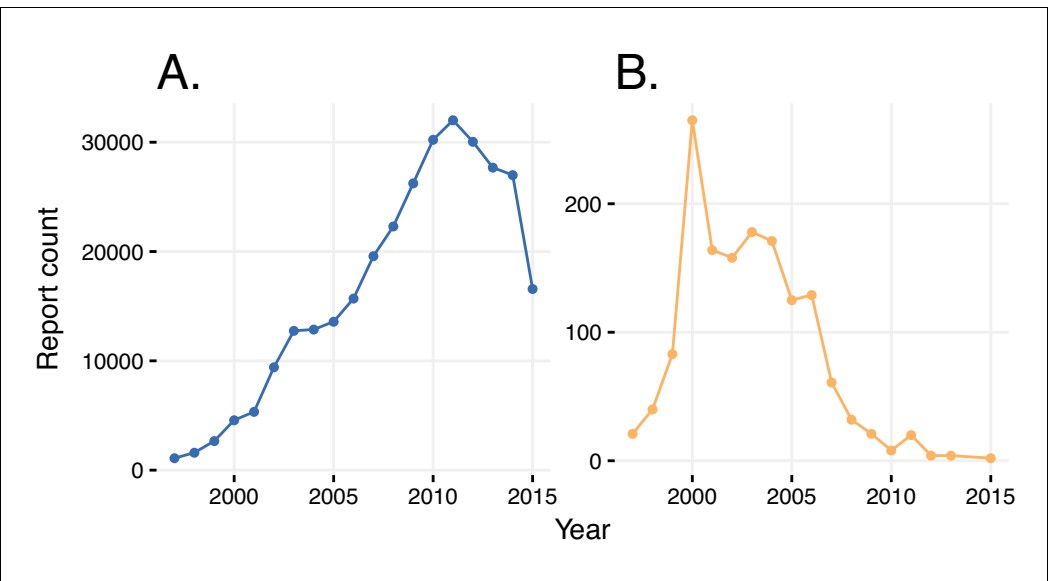

**Figure 3.** Reports with wrongly identified indications or ADRs. (**A**) Total number of reports in a given year where the same indication and ADR were reported. (**B**) Number of reports in a given year where diabetes was stated as the adverse reaction caused by rosiglitazone.

Myocardial infarction (RRR = 17.85; q-value $<10^{-5}$) and cerebrovascular accident (RRR = 17.69; q-value $<10^{-5}$) accounted for a large proportion of the ADRs reported for rofecoxib from its introduction in 1999 (*Figure 4B and D*). Before the study by *Bombardier et al. (2000)*, most reports were filed by physicians. Between the Bombardier publication and the introduction of the label warning, these physician reports remained constant, while the number of reports by lawyers grew substantially. After the introduction of the label warning, the number of reports from physicians slightly decreased, but the trend to attribute myocardial infarction and cerebrovascular accident to administration of rofecoxib was further cemented by submitters who identified themselves as lawyers (see *Figure 4C*).

We also inspected the time evolution of another COX-2 inhibiting NSAID, celecoxib, approved by the FDA in December 1998, just shortly before Vioxx (*Figure 5*). Inspection of the timeline of celecoxib reports shows a slight increase in the number of reports around September 2004, reflecting the increase in use associated with the withdrawal of rofecoxib (*Figure 5A*). Until December 2004, the pattern of ADR in celecoxib reports is dominated by cerebrovascular accident (per-month RRR up to ~35) and myocardial infarction (per-month RRR up to ~45) in a similar fashion as in rofecoxib reports (*Figure 5B* and *Figure 5D*). The increase of the overall number of reports around September 2004 coincided with concerns about the safety of celecoxib, likely reflecting a report of increased risk of cardiovascular events in patients who used celecoxib systematically over prolonged periods of time (*Solomon et al., 2005*). We checked whether the trends in reporting of side effects of celecoxib was affected by co-administration of rofecoxib, but the distribution of ADRs was almost identical after excluding the 8% of reports in which rofecoxib was present as a concomitant drug (*Figure 5C*). Closer examination of this pattern revealed that the reports during this period of time were largely submitted by lawyers and 'unidentified' individuals, while the contribution of health professionals remained steady much below the level of reports for rofecoxib (*Figure 5E*). These trends were confirmed by logistic regression modeling (see Materials and methods and *Supplementary file 2*), which showed that reports of myocardial infarction were significantly correlated with reports of celecoxib filed by lawyers before 2005 (Model four in *Supplementary file 2*). We also considered the Pfizer moratorium on direct-to-consumer advertising in 2004 as a significant event on sales and consequently ADR reporting (*Consumers Union and Consumer Reports Best Buy Drugs, 2005*). After the Vioxx case, the FDA issued new labeling for not just COX-2 selective but for all NSAIDs which set new safety standards for the anti-inflammatory arena (*U.S. Food and Drug*

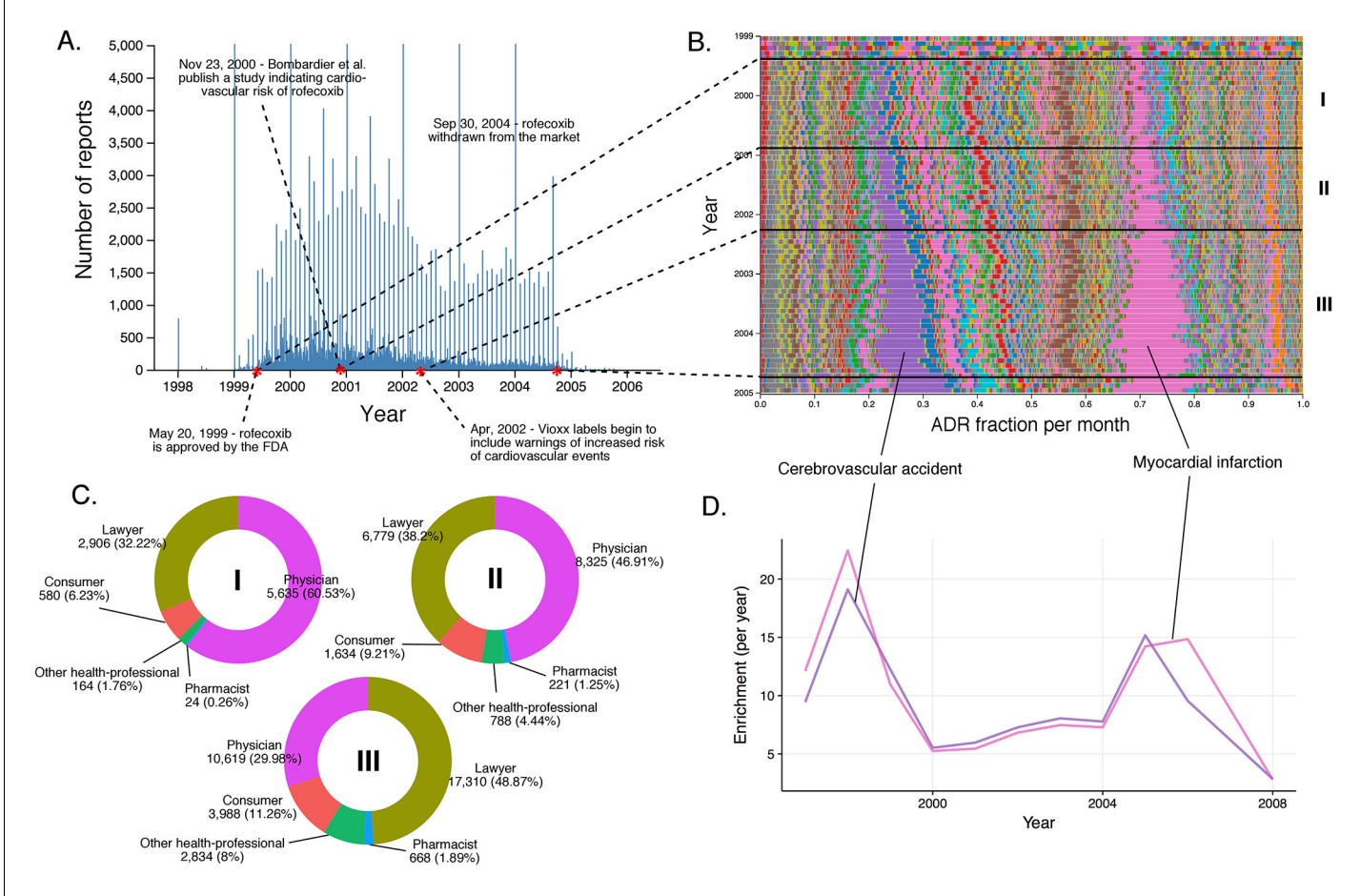

**Figure 4.** Submission pattern and time evolution of rofecoxib FAERS reports. (**A**) Number of reports (per day) where rofecoxib was reported as primary suspect. Red dots represent events with a major impact on the FAERS reporting pattern of rofecoxib. (**B**) Relative percent participation of all 'preferred term' (PT)-level ADRs observed for rofecoxib. Each ADR is represented by a separate color. Characteristic time periods on the timeline of this drug are demarked by lines (associated with definitive events), and numbered. Monthly ADR fractions shown here are also reported in *Supplementary file 1*. (**C**) Identities of those reporting rofecoxib ADRs at the various reporting periods, marked to correspond with the Roman numeral annotations in panel B. (**D**) Enrichment-based clusters of ADRs (cerebrovascular accident and myocardial infarction) observed in rofecoxib reports between 1997 and 2006.

*Administration, 2016*). While Vioxx was removed from the clinic, Celebrex enjoyed a revival after 2004, in particular as the only COX-2 selective NSAID in the clinic and because of the updated labels for other NSAIDs. Pfizer resumed careful advertising with clear reference to side effects that helped the recovery of clinical celecoxib use. While we examined these stimulating effects, we also recognized that the sales volume was likely to grow but we were not able to determine the volume because of coincidental price increase interference (*Schondelmeyer and Purvis, 2014*). This case clearly demonstrates the complexity of the performance of drugs in the post-marketing environment where ADR reporting could be significantly modified by multiple factors, including regulatory and social aspects.

## Drugs with similar chemical structure and modes of action may display distinct clinical ADR phenotypes

It is generally expected that compounds with similar structures and modes of action will have similar ADR profiles; for instance, several selective serotonin reuptake inhibitors (SSRIs) are associated with suicidal behavior in young adults (*U.S. Food and Drug Administration, 2016*; *Muller et al., 2015*). However, this is not always the case. The post-marketing ADR reports of the structural analogs rosiglitazone (*Greene, 1999*) and pioglitazone, which act on the same primary target peroxisome

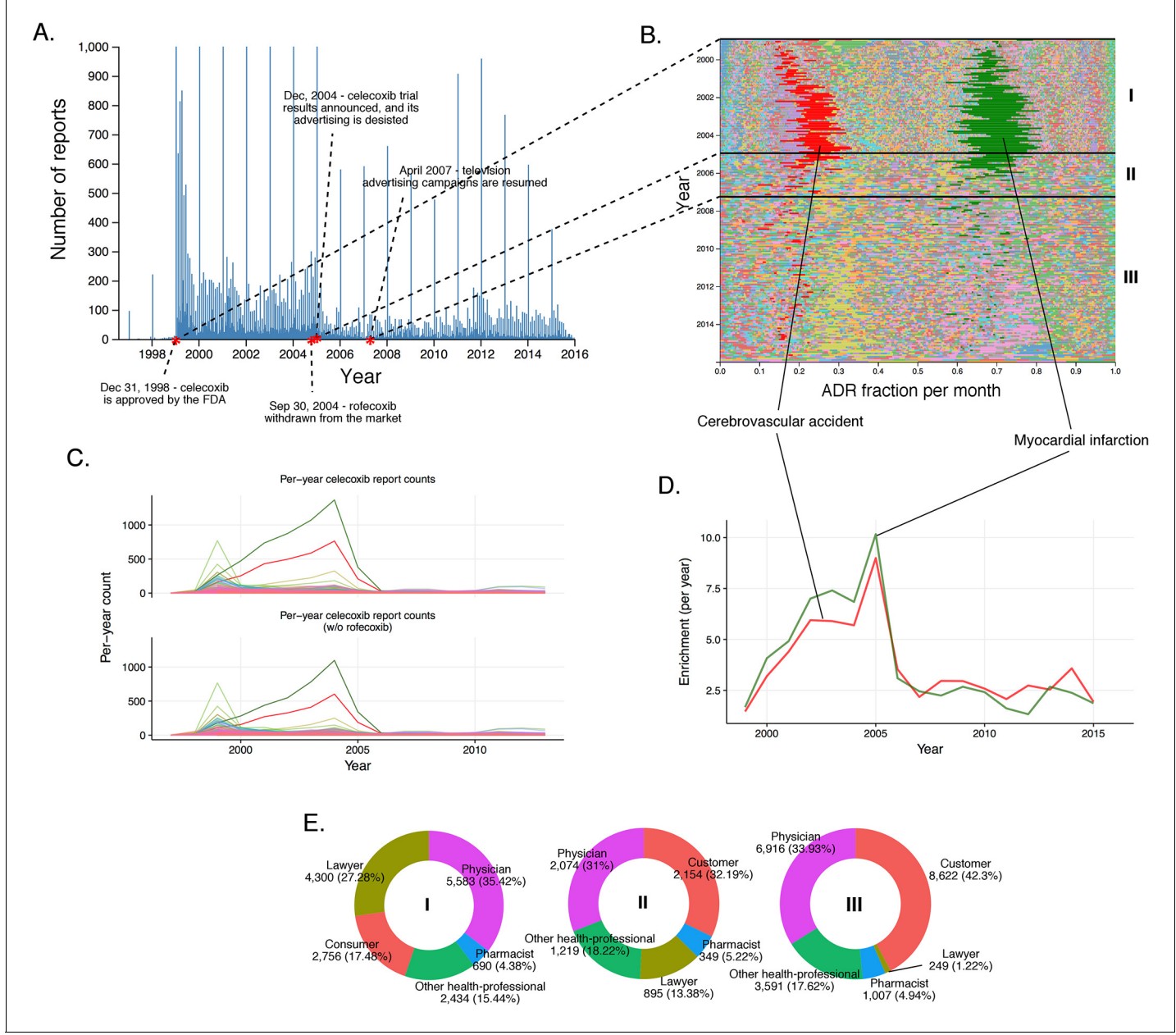

**Figure 5.** History of FAERS reports on celecoxib. (**A**) Number of FAERS reports (per day) where celecoxib was reported as primary suspect. (**B**) Relative percent participation of all PT-level ADRs observed for celecoxib. Each ADR is represented by a separate color. Characteristic time periods on the timeline of this drug are marked by lines, and numbered. Monthly ADR fractions shown here are also reported in *Supplementary file 1*. (**C**) Per-month number of reports where celecoxib was primary suspect; each line corresponds to a separate PT-level ADR. The top plot describes all reports with celecoxib as primary suspect. In the plot on the bottom the reports in which rofecoxib was also present were omitted. Colors are matched with those used in panel B. (**D**) Enrichment-based clusters of most frequently reported ADRs (cerebrovascular accident and myocardial infarction) observed in ccoxib reports. Colors match those in B and C. Note, that this plot will not exactly correspond to panel B, because enrichments presented here show the ratio of the number of observed events in a given year compared to what one would expect at random, while the traces in B show a proportion of a given ADR compared to other ADRs during a given period of time. (**E**) Identities of those reporting celecoxib ADRs at various reporting periods, marked to correspond with the Roman numeral annotations in panel B.

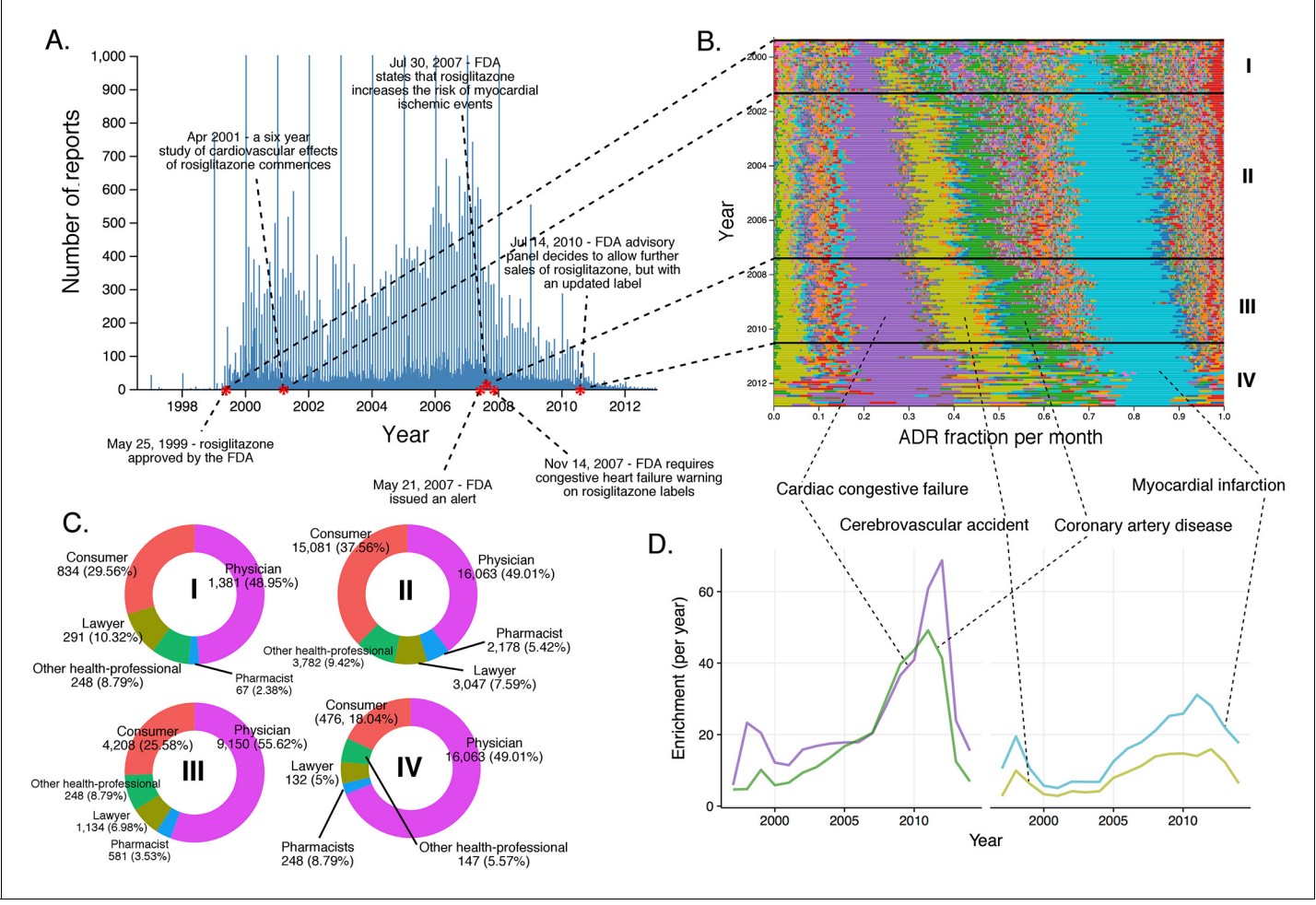

**Figure 6.** Rosiglitazone reports. (**A**) Number of FAERS reports (per day) where rosiglitazone was reported as primary suspect. (**B**) Per-month percent participation of all PT-level ADRs observed for rosiglitazone. Each ADR is represented by a separate color. Characteristic time periods on the timeline of this drug are demarked by lines, and numbered. Monthly ADR fractions shown here are also reported in *Supplementary file 1*. (**C**) Identities of those reporting rosiglitazone ADRs at various reporting periods, marked to correspond with the Roman numeral annotations in panel B. (**D**) Enrichment-based clusters of ADRs observed in rosiglitazone reports.

proliferator activated receptor γ (PPAR-γ) and are structurally related, are notably different (compare *Figures 6B* and *7B*).

For rosiglitazone, many heart-related reports have been filed since its FDA approval in May 1999 (*Figure 6A*, *Figure 6B*). Whereas the absolute number of reports have varied over time, and has been affected by the clinical trial and scientific reports in much the same way as rofecoxib, the predominance of heart effects, such as congestive cardiac failure (RRR = 31.99; q-value <$10^{-5}$), coronary artery disease (RRR = 26.32; q-value <$10^{-5}$), cerebrovascular accident (RRR = 11.72; q-value <$10^{-5}$), and myocardial infarction (RRR = 20.73; q-value <$10^{-5}$), relative to other events, has been unperturbed throughout the lifetime of this drug (*Figure 6B and D*).

The other hypoglycemic drug, pioglitazone, has triggered fewer reports of heart effects relative to the clinical ADR profile of rosiglitazone since its approval in July 1999 (*Figure 7A*). Although analysis of FAERS reports does support a statistically significant signal between pioglitazone and cardiac failure (RRR = 5.09; q-value <$10^{-5}$), the time evolution of this signal reveals that the major contribution to its statistical strength comes from a single peak that subsides by the year 2002, and coincides with the increased scrutiny of rosiglitazone (*Figure 7C*). Unlike rosiglitazone, the ADR landscape of pioglitazone is dominated by bladder cancer (RRR = 305.69; q-value <$10^{-5}$), with a substantial increase in reports from 2009 onward (*Figure 7B*). Conversely, this signal is significantly

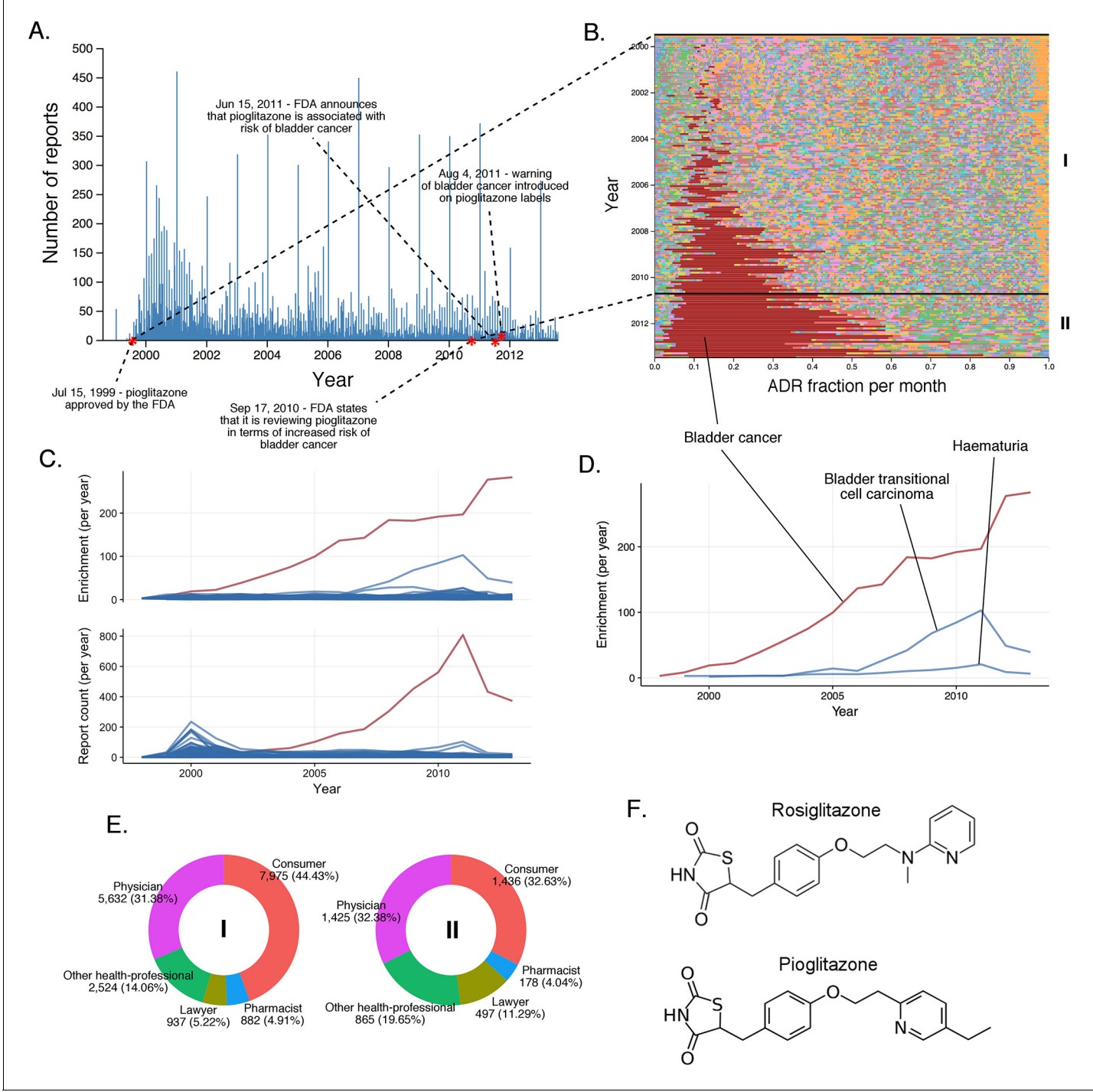

**Figure 7.** The landscape of pioglitazone reports. (A) Number of FAERS reports (per day) where pioglitazone was reported as primary suspect. (B) Per-month percent participation of all PT-level ADRs was observed for pioglitazone. Each ADR is represented by a separate color. Characteristic time periods on the timeline of this drug are marked by lines and numbered. The monthly ADR fractions shown here are also reported in *Supplementary file 1*. (C) Per-month number of reports where pioglitazone was primary suspect; each line corresponds to a separate PT-level ADR. The plot on the top of the panel shows number of times individual ADRs have been reported, and the bottom the corresponding per-month enrichments. The traces for cardiac failure have been distinguished by the blue color. (D) Enrichment-based clusters of cancer-related ADRs observed in pioglitazone reports. (E) Identities of those reporting pioglitazone ADRs at various reporting periods, marked to correspond with the Roman numeral annotations in panel B. (F) Structure of rosiglitazone and pioglitazone.

underrepresented in the rosiglitazone reports (RRR = 0.12; q-value $<10^{-5}$). There is evidence that non-selective PPAR agonists (α + γ) such as pioglitazone could contribute to carcinogenesis (*Oleksiewicz et al., 2008*; *Piccinni et al., 2011*), and a recent study linked bladder cancer to the development of chronic kidney disease as an effect of long-term use of pioglitazone (*Lee et al., 2014*). This observation was not confirmed with short-term use of pioglitazone (*Lewis et al., 2011*), and it is important to note that the increase in incidence of bladder cancer is linked to pioglitazone treatment that is ongoing for more than 2 years. Aside from the scientific evidence, it is clear from the submitter population (mostly lawyers and consumers; see *Figure 7*) that bladder cancer report-ing has a significant stimulated component in FAERS. As there is no clear etiology for the high inci-dence of this ADR, further investigation is needed to explore the effect of concomitant therapies such as glucocorticoids which might influence the side effect profile of pioglitazone. Still, the mecha-nisms linking the less selective pioglitazone but not the selective PPAR-γ agonist rosiglitazone to bladder cancer are unclear, and this association must remain tentative.

## Using monthly report counts to de-bias stimulated reporting

The trends and biases in ADR reporting can hamper the division and reliability of drug-ADR associa-tions. The statistically significant association that we found between pioglitazone and cardiac failure stems mostly from the reports from before 2004, which may reflect the popular view that hypoglyce-mic thiazolidinediones cause cardiovascular side effects (*Nissen and Wolski, 2007*). Whereas we cannot discount that this reflects genuine events, a time-resolved statistical analysis tilts against this. In a month-resolved statistical analysis of the significance of the pioglitazone – cardiac failure associ-ation, most dates indicated that there was no statistically significant association between this drug-ADR pair (top panel in *Figure 8*). Conversely, the association of rosiglitazone and myocardial infarc-tion was statistically significant in nearly every time period (bottom panel in *Figure 8*). The periods where the pioglitazone - cardiac failure association is statistically significant are restricted to a couple of sparse spikes (*Figure 8*), and so we suggest this association to be stimulated, and most likely arte-factual. Such month-resolved statistical analysis for drug-ADR associations may be broadly helpful in detecting biased reporting trends. Continuing observation of changes in reporting patterns and cor-relative studies can reveal further aspects of the potential discrepancies between the clinical profiles of structurally similar drugs. Published data on the favorable effects of pioglitazone on lipids, primar-ily on triglycerides, over rosiglitazone also supports our observations (*Nissen and Wolski, 2007*; *Goldberg et al., 2005*; *Lincoff et al., 2007*).

## Combining pharmacokinetics and FAERS to investigate mechanism and for reverse translation

There is a great interest in using pharmacovigilance for target identification and to illuminate thera-peutic and ADR mechanism of action (*Muller et al., 2015*; *Nguyen and Lewis, 2014*; *Rothman et al., 2000*; *Shively et al., 1999*; *Urban et al., 2014*). By matching to in vitro activity, one may hope to associate an ADR that emerges in FAERS with the targets responsible for the physiol-ogy, making the linkage: drug → known target → ADR. Whereas we ourselves have championed the role of in vitro pharmacology for anticipating possible toxicology (*Lounkine et al., 2012*; *Bowes et al., 2012*; *Bender et al., 2007*), doing this reliably depends on knowing the exposure of the drug to the implicated target. Without considering drug pharmacokinetics, FAERS-based infer-ence of drug → target → ADR associations can mislead (*Muller and Milton, 2012*).

An illustrative example is the hypertension associated with inhibition of the vascular endothelial growth factor or its receptor (VEGF, VEGF-R2; see Materials and methods). The relevance of such inhibition to hypertension is supported by the high-incidence of this ADR with the VEGF-R2-specific humanized antibody, bevacizumab (*Figure 9B*) (*Zhu et al., 2007*). Correspondingly, several small molecule kinase inhibitors that inhibit VEGF-R2 with relevant in vivo pharmacokinetics (*Figure 9B*) also share the hypertension side effect. However, other kinase inhibitors with VEGF-R2 inhibition do not appear to increase reports of hypertension (*Figure 9*). High incidence is reported only with those drugs in this class that have exposure margins (EM) less than 13 for this target (biochemical $IC_{50}$/ $C_{max}$; *Figure 9*). We considered several caveats in this analysis: (1) human exposures could range 5–10 fold, (2) many of the clinically approved kinase inhibitors are promiscuous with a few targets which might affect blood pressure, and (3) in some instances, the clinical sample size is relatively

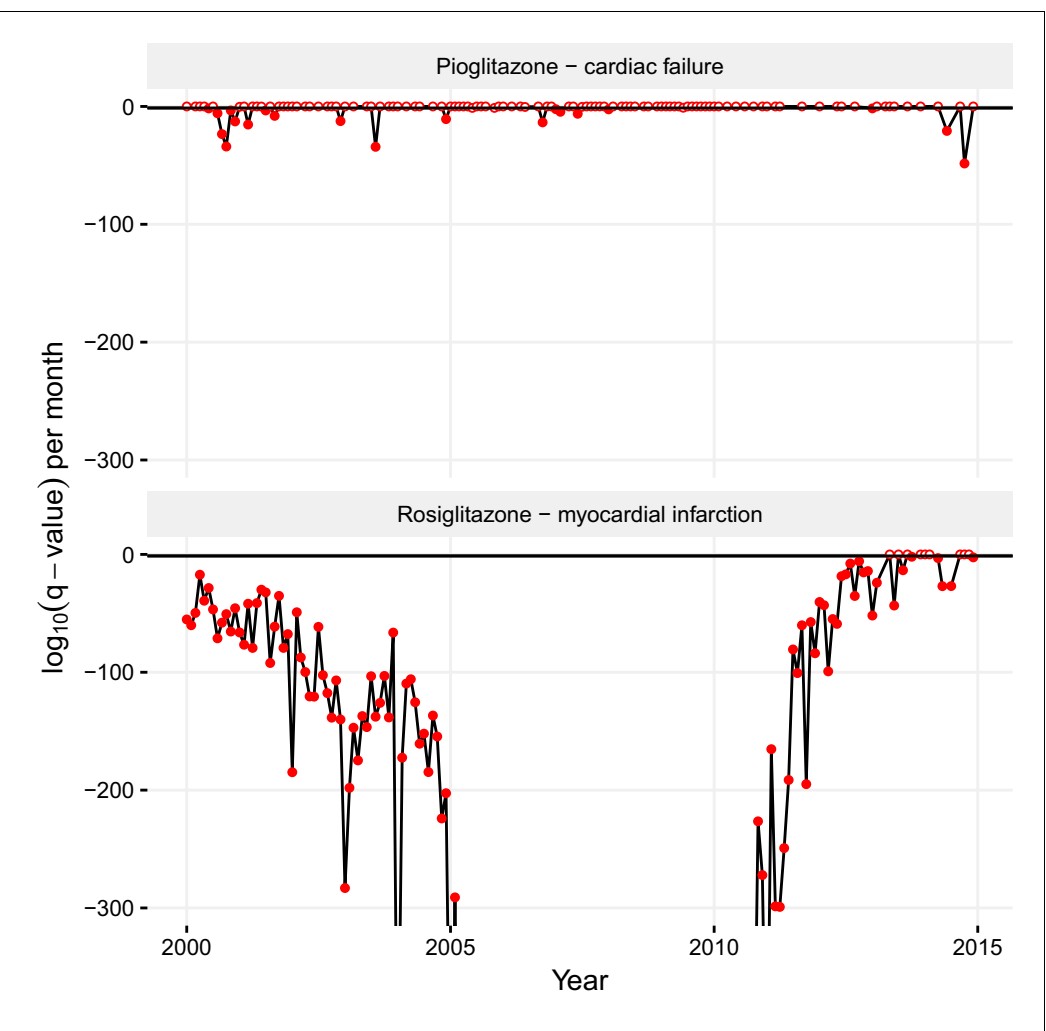

**Figure 8.** Statistical significance of association between pioglitazone and cardiac failure (top panel), and rosiglitazone and myocardial infarction (lower panel) over time. The horizontal line demarks the critical q-value cutoff of 0.05, below which the association becomes statistically significant. On dates when the association crosses this threshold, its q-value is indicated by a filled circle; otherwise, it is indicated by an empty circle. The extreme q-values below $1 \cdot 10^{-300}$ are not shown.

small. While all these factors could influence the incidence of hypertension associated with VEGF-R inhibition, it is possible to recommend an exposure margin around 10 for clinical candidates in pre-clinical development to avoid this side effect in the clinic (*Figure 9A*). Thus, using such an EM cutoff in the FAERS analysis, the signal for this ADR over random will separate drugs with true adverse event from those that lack it (*Figure 9*).

A more complex case emerges through the investigation of methylphenidate and the atypical antipsychotics, risperidone/paliperidone (*Corena-McLeod, 2015*) and aripiprazole, drugs prescribed for the treatment of attention deficit hyperactivity disorder (ADHD) (*Correia Filho et al., 2005*; *Ercan et al., 2012*; *Fernández-Mayoralas et al., 2012*). FAERS analysis indicates that treatment with risperidone/paliperidone, the latter of which is the main active metabolite of risperidone, increases the frequency of gynecomastia and galactorrhea, while methylphenidate has a low incidence of these ADRs, as do other atypical antipsychotics, such as aripiprazole (*Figure 10*) (*RELX Intellectual Properties SA, 2016*). For example, between 2007 and 2013, there were 5073 and 123 cases in FAERS, respectively, where risperidone and paliperidone are the primary suspect of gynecomastia (*Figure 10*, RRR = 113.82, q-value $<10^{-5}$; RRR = 7.53, q-value $<10^{-5}$). For

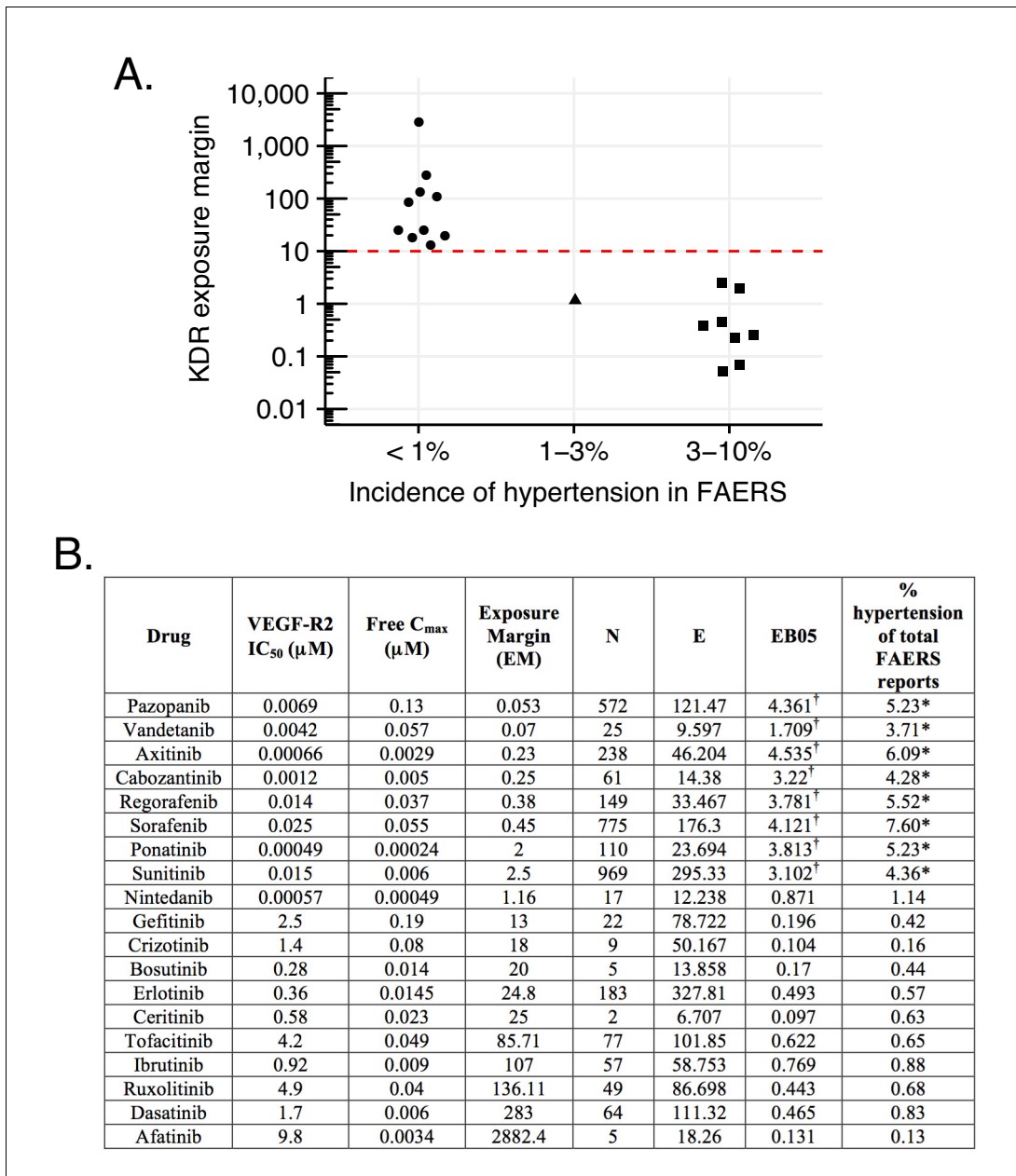

**Figure 9.** Hypertension associated with VEGF-R2 inhibition depends on the exposure margin of small molecule anti-VEGF-R2 drugs (VEGF-R2 $IC_{50}/C_{max}$). (**A**) Suggested exposure margin for marketed VEGF-R inhibitors based on post-marketing incidence of hypertension in correlation with plasma exposure (VEGF-R $IC_{50}$/free $C_{max}$). The proposed 10 times margin represents clear separation of VEGF-R inhibitors with and without significant increase in hypertension with the only exception of nintedanib. (**B**) FAERS reports of small molecule kinase inhibitors with VEGF-R2 inhibition show an increased incidence of hypertension reports only in case their exposure margin is less than 13. The label of drugs with high incidence of hypertension in FAERS lists this side effect, while none of those drugs that have low incidence carry the label. *p-value of association between drug and hypertension <0.001. Counts (**N**), expected counts (**E**), and an often-used disproportionality measure (EB05) based on the FDA's FAERS database of spontaneous reports of suspected drug adverse drug reactions are provided. The values of E are the expected number of patients reporting vascular hypertensive disorder after taking each drug if the drug reports and the reports of the event were independent within the database, conditional on the patients age and gender. The ratio N/E is a measure of disproportionality of report counts of each particular drug-event combination. The value EB05 (empirical Bayes 5% lower bound of a 90% credible interval) is a conservative estimate of the true reporting disproportionality that uses estimated overall prevalence of drug-ADR associations throughout the

*Figure 9 continued on next page*

*Figure 9 continued*

database. The value of EB05 is less than N/E and has the effect of correcting the simple ratio for sampling variance and multiple comparisons bias. See literature (***DuMouchel, 1999***; ***DuMouchel and Pregibon, 2001***; ***Szarfman et al., 2002***; ***Almenoff et al., 2007***) for details and discussion of the FAERS database and the use of disproportionality analyses within spontaneous report databases. The values of EB05 for the first three drugs indicate 95% confidence that reports of those three drug-event combinations are reported about three or four times as often as would be expected if they were independent, while the values of EB05 <1 in the final three drugs in the table indicate no evidence for higher than expected reporting rates. More detailed results from Bayesian analysis are available in ***Supplementary file 3***. [†]Significant increase.

aripiprazole (RRR = 0.85) and methylphenidate (RRR = 1.39), however, the q-values were close to 1, indicating no significant associations with this ADR (***Figure 10B***). Thus, the FAERS data clearly separates the profile of risperidone/paliperidone from both methylphenidate, with which it overlaps for treatment of ADHD, and from other atypical antipsychotics, like aripiprazole. The inference would be that the target responsible for the gynecomastia and galactorrhea for risperidone/paliperidone is not modulated by either methylphenidate or any other atypical antipsychotics. Although this is correct for methylphenidate, it is incorrect for the atypical antipsychotics. The clinical profile of atypical antipsychotic drugs depends on their pattern of engagement with central nervous system targets largely receptors and transporters (***Richelson, 1996***).

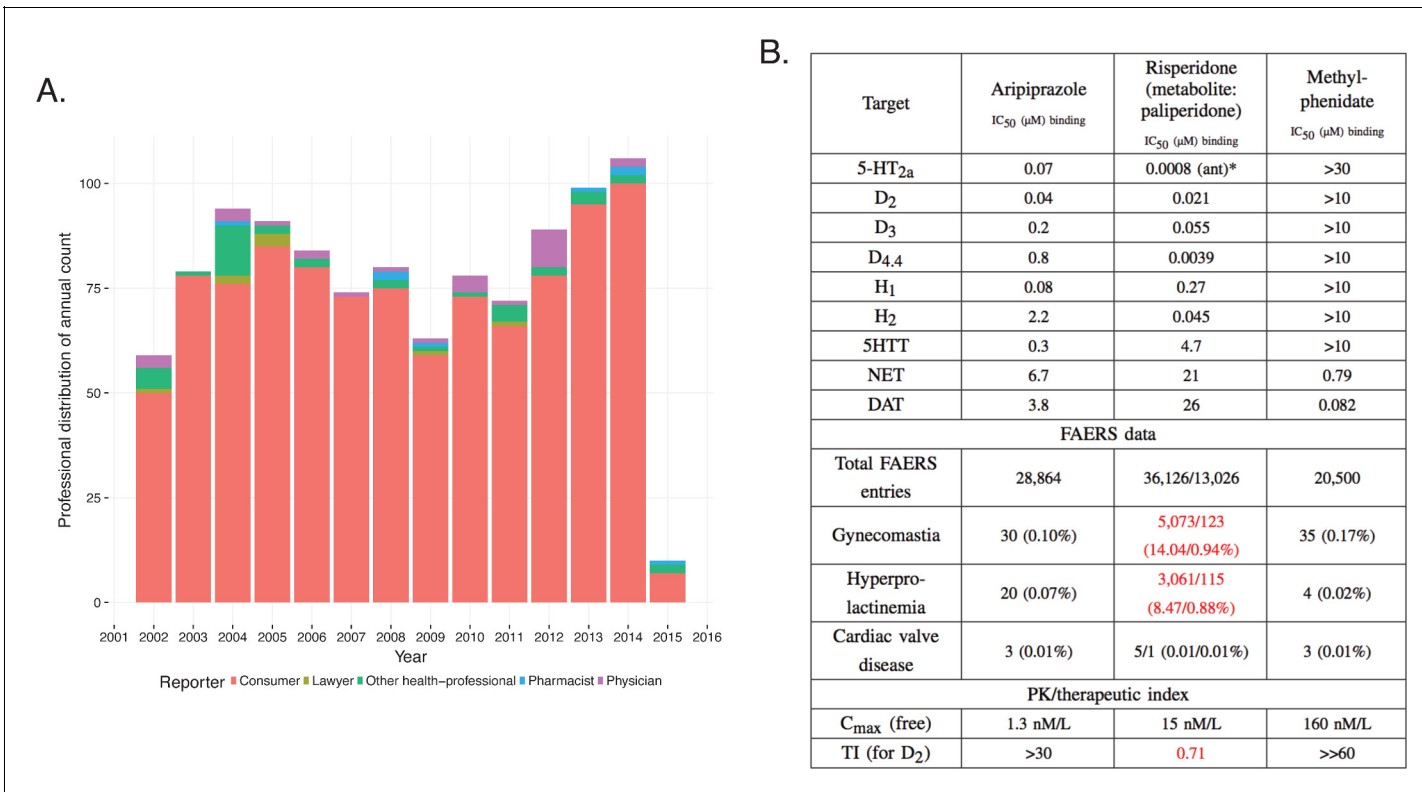

**Figure 10.** Integration of pharmacodynamic and pharmacokinetic data is necessary to interpret FAERS information. (A) FAERS analysis of the reporting pattern of gynecomastia in patients treated with risperidone between 2002–2015. (B) Summary table of the in vitro pharmacological profile, FAERS entries (total number of reports, and reports of gynecomastia, hyperprolactinemia and cardiac valve disease where the listed drugs were the primary suspects) and calculation of exposure margin of aripiprazole, risperidone/paliperidone and methylphenidate. The prominent effects of risperidone/paliperidone at the $D_2$ dopamine receptor in conjunction of the narrow TI differentiates these compound(s) from the rest. Assays were performed at the Novartis Institutes for BioMedical Research, Cambridge. *Asterisks denote functional assays. ant: antagonism.

Both aripiprazole and risperidone/paliperidone are atypical antipsychotics with high affinity to dopaminergic, serotonergic, adrenergic, and histaminergic receptors (Figure 10B [*Lounkine et al., 2012*; *Roth et al., 2000*]). It is well established that inhibition of the $D_2$ dopamine but not 5-$HT_{2C}$ receptors are linked to hyperprolactinemia, which is the underlying mechanism of gynecomastia and galactorrhea (*Calarge et al., 2009*; *Alladi et al., 2017*). Regardless of their similar potency at the $D_2$ dopamine receptor the difference between the ADR profile of aripiprazole and risperidone/paliperidone is explained by their mechanism of action (risperidone is a full antagonist and aripiprazole is a partial agonist) and particularly by their PK profile, which reveals that the exposure margin (EM [*Muller and Milton, 2012*]) for $D_2$ is large for aripiprazole and so this ADR did not manifest. For risperidone, the EM is less than one which explains the high incidence of gynecomastia (*Figure 10*). Methylphenidate does not affect the $D_2$ receptor at all, and accordingly this ADR was not observed.

While paliperidone is known to cause hyperprolactinemia (*Bostwick et al., 2009*) which is confirmed by the FAERS data, it remains to be explained why it has a significantly lower reporting rate than its parent, risperidone, with similar activity at the $D_2$ receptor (*Arakawa et al., 2008*).

## Discussion

Four key observations emerge from this study. *First*, much of the potential signal in FAERS and related databases is obscured by chemical name redundancy. This introduces false associations that would fall to insignificance on synonym aggregation, and this hides associations that would be significant on aggregation. This may be addressed by representing active ingredients by their unique chemical structures in a readily searchable form as we do here (*Supplementary file 1*). *Second*, FAERS reports tilt toward serious outcomes, partly owing to a confusion of ADRs and outcomes. *Third*, FAERS suffers from several forms of conflation: multiple entries, indications with ADRs, newsworthiness, and scientific and legal influences. These may be detected by statistical analyses, including comparing reports over time. *Fourth*, and perhaps more generatively, once these biases and conflations are corrected, the molecular mechanism of previously hidden ADRs can be revealed; an example explored here is the association of urinary bladder cancer with mixed PPAR-$\alpha$ and PPAR-$\gamma$ agonists.

A major reason for FAERS's bias toward serious outcomes is the conflation of ADRs and outcomes. This may stem from an issue as simple as confusion on whether 'death' is listed as an ADR – associated with the drug only – or an outcome – associated with the disease itself. This is the case with the attribution of the side effect 'death' to thalidomide's use in complex myeloma multiplex, when this reflects the high mortality rate of the disease itself (*Greene, 1999*). Naturally, there are some cases where use of a drug can increase death rate, even in treating life-threatening diseases, such as the case of milrinone for acute heart failure syndromes (AHFS) (*Bayram et al., 2005*) or severe chronic heart failure (*Packer et al., 1991*). Matters can be improved when a drug is used in different indications with distinct symptoms and outcomes, enabling differentiation between disease- and drug-related outcomes. FAERS could be further developed to automatically alert the investigator to common indication biases, such as high death rate in malignancies, or baseline metabolic anomalies in diabetes. Careful statistical analysis is needed in these cases to differentiate between the outcomes associated with the disease or with the suspect drug. For now the category 'outcome' should be used cautiously for ADR analysis, especially in large-scale studies that aggregate data from several drugs.

In principle, submission of FAERS reports requires medical knowledge, as they include specific indications for which drugs were prescribed, identification of the primary suspect of ADRs, and structured description of ADRs by MedDRA terms. Nevertheless, a third of the reports are contributed by customers, and a half by submitters who do not identify themselves as medical professionals, including lawyers. This contributes to the high redundancy and error in FAERS, and to the 'stimulated reporting' from which it suffers (*Hoffman et al., 2014*). This appears to have been the case with celecoxib, whose association with cerebro- and cardiovascular events in FAERS reports was driven primarily by reports from legal professionals (*Figure 5E*). After rofecoxib was withdrawn, the proportion of these events for celecoxib returned to background. For cases like these, a temporal analysis of ADR-drug associations is essential as it can pinpoint spurious associations. Interrogation of FAERS and related databases to illuminate the molecular mechanisms of ADRs, and indeed the shared target profiles of drugs, has been an area of much recent interest (*Center for Drug*

*Evaluation and Research, 2015*). Here, too, we find that the disambiguation of ADRs, indications, reporting and indication biases can reveal previously obscured associations. An example is the association of bladder cancer with the mixed PPAR-α and PPAR-γ agonist pioglitazone. FAERS analysis is instrumental here, providing information on a large patient population and enabling the comparison with the selective PPAR-γ agonist rosiglitazone, which is not associated with bladder cancer (*Figure 6*) (*Smith, 2001*).

Improvement of statistical methods for signal detection is an area of active research (*Wysowski and Swartz, 2005*; *Lasser et al., 2002*; *Friedman et al., 1999*; *Moore et al., 1998*; *Harpaz et al., 2013a*) and much attention is paid to advanced statistical methods such as (Bayesian) information components (*Harpaz et al., 2013a*), Empirical Bayes statistics (*Harpaz et al., 2013b*), and hierarchical methods (*Harpaz et al., 2013b*). As with all machine learning and statistical approaches, these methods assume clean input data – the biases and noise they address is of statistical nature. We have used a well-known disproportionality approach, relative reporting ratio (RRR) with $\chi^2$ test statistic for disproportionality. The RRR has its limitations and may underperform compared to more advanced methods (*Harpaz et al., 2013a*). The focus of our study was how proper preparation of the input data – cleaning drug ingredient mapping, and estimating multiple reporting – boosts signal detection performance, even with a simple method such as the RRR. We believe that applying the procedures and precautions we described here together with more advanced statistical methods will boost their performance even further.

A key caution is that simple associations such as drug-ADR or drug-target do not yield clinical relevance without pharmacokinetic information, which ensures that the implicated target is exposed to the drug at effective concentrations. This is illustrated by the comparison of the ADHD drugs risperidone and aripiprazole and gynecomastia. Both drugs affect the $D_2$ dopamine receptor that underlies the ADR, but only risperidone reaches a sufficient exposure to trigger it. The VEGF-R2 inhibitor example suggests that this type of evaluation is the only way one can objectively detect ADR-target pairs and explain the underlying mechanisms of their manifestation. Relying only on a ADR → drug → in vitro target schema can be insufficient to understand shared targets or molecular mechanisms; as Goodman long ago suggested, pharmacokinetic exposure remains crucial (*Goodman and Gilman, 1985*).

FAERS, an already valuable asset for clinicians and pharmaceutical scientists, could be improved in several ways to improve post-marketing pharmacovigilance. First, we recommend introducing automatic mapping of drugs and synonyms to ingredients, as discussed in this paper. Alerts could be issued for indications where serious outcomes are common and hard to distinguish from ADRs. Definitive drug-ADR associations would require information on exposure. Here, the availability of applied dose and associated PK data are essential. At present, no dose is provided and PK data can be obtained only from different sources. One of such sources is PharmaPendium (*RELX Intellectual Properties SA, 2016*), which contains both FAERS data and PK information, but even here the data are not linked directly, and this resource is only commercially and not publicly available. Public databases to which FAERS could link include DailyMed or drugs.com, which would potentially provide information on pharmacokinetics, drug labels, formulations, and approved indications.

There are several weaknesses of the present approach, which seems to be more or less general for mining efforts of FAERS. We encountered difficulties in differentiating drug-ADR associations when several drugs were co-administered, particularly when combinations contain drugs with possible synergistic or antagonistic effects. Also, different formulations of the same active drug ingredient could contribute to differences of the drug's profile, which is difficult to capture with the present approach.

Finally, the introduction of reporting automation into FAERS would serve the dual purpose of reducing errors, such misclassifying ADRs and indications, and making the tool more interactive and rewarding for health professionals. One can imagine submitters receiving feedback on similar entries, including cases on the same suspect drug, indication, patient population and most common or problematic treatment regimens. The tool could also help to define the 'suspect drug' in treatment scenarios, independent from the submitters' intention, among other advantages. While this development would need investment, one could imagine undertaking it as part of a public-private partnership from which all would benefit.

## Conclusions

The challenges and opportunities in FAERS and indeed from related databases flow from its ambitions. It publishes multiple reports - physicians, patients, other medical professionals, attorneys - on multiple drugs, named in multiple ways and taken in multiple contexts. FAERS does not represent a strictly reviewed and carefully channeled source of observations about drugs, as a clinical trial does - there is no placebo arm in FAERS, nor are there reports of cases when a given drug was prescribed and caused no side effects. It contains uncontrolled, volunteered information on a large scale. This may be seen as a feature of FAERS - a database designed with hypothesis generation rather than hypothesis testing in mind. Still, the hypotheses that FAERS suggests depend critically on the ability to disentangle its data. Tools like those described here are crucial to control for the often conflated and contradictory observations in FAERS reports, where serious outcomes are over-reported, reported death is often linked to submission by the patients themselves, a single event is reported multiple times, true associations between drugs and adverse events are missed because a single agent is named in multiple ways, or a mechanistically related disease occurs in different system organ categories. Once its data are disentangled, FAERS represents unprecedented opportunities to track drug outcomes in large patient populations, revealing new associations. The power of such analysis is that it may be applied systematically and comprehensively across a massive number of observations.

We recognize that a fully automated method, such as that described here, cannot replace expert knowledge. What such a method can do is identify, prioritize and sometimes deprioritize drug-adverse event associations, and sometimes even mechanistic inference, for detailed expert identification. This approach should be useful to the growing community of regulators, payers, physicians, and patients that work with and depend upon trends emerging in FAERS to improve drug use and health outcomes. By making several of these tools available to the community, we hope to enable future interrogation of FAERS by other investigators.

## Materials and methods

### FAERS data source

FAERS reports were downloaded on May 24th 2016 from the FEARS database (*U.S. Food and Drug Administration, 2016*) for the years between fourth quarter of 1997 and fourth quarter of 2015, inclusive. ADR, indication, drug role (primary suspect, secondary suspect, concomitant), and outcome data was mapped using ISR report identifiers to the individual reports. Drugs were identified by the reported drug name in FAERS.

### Mapping drugs to ingredients

We assembled a list of synonyms of drugs, using public and licensed databases including Thompson Reuters Integrity (*Thomson Reuters, 2016*), GVK (*GVK Biosciences, 2014*), Drugbank (*Law et al., 2014*), ChEMBL (*Gaulton et al., 2012*), and R$_x$Norm (*Nelson et al., 2011*). These synonyms were matched with drug products, and constituent molecular ingredient structures, encoded using as InChIKeys (*International Union of Pure and Applied Chemistry, 2016*). We read in all the drug names from all the FAERS reports, and all the synonyms that had been assembled. Non-alphabetical characters (except numbers), capitalization, and terms that carried little information regarding the identity of the drugs (such as articles, or often occurring words like 'acid') were removed from the FAERS drug names and synonym names, and the remaining parts of the names were tokenized. Each tokenized FAERS drug was then compared to each tokenized synonym, and the overlap of tokens was recorded for each pair using the Tanimoto similarity coefficient $t_c$. The synonym with the highest $t_c$ value was picked for a given drug, as long as the $t_c$ was $\geq 0.2$; for any drug, if a synonym with $t_c$ of 0.99 or higher was found, it was considered to be an exact match, and used to identify the drug in question without comparison to further synonyms. For the most frequent (among the top 500) drug names in FAERS, we manually mapped those drug names to InChIKeys that could not be mapped. Since InChIKeys are not typically calculated for large macromolecules, we used the non-proprietary name in lieu of the InChIKey in these cases.

## Adverse drug reaction terms

The majority of ADRs in FAERS are reported using the Medical Dictionary for Regulatory Activities (MedDRA) (*Brown et al., 1999*). Some older reports contained terms that are not part of the newer MedDRA that is used currently. To normalize and annotate the ADR terms extracted from reports we used a Levenshtein algorithm that compared the FAERS ADR terms to the MedDRA terminology. We set the minimal Levenshtein score at which a given MedDRA term was considered a perfect match to 0.95, and the minimal acceptable score to 0.90 at above which the highest scoring term was picked to standardize a given ADR. Additional 32 ADR terms were standardized manually, leaving less than 0.5% ADR terms unmatched.

## Establishing ingredient - ADR and ingredient - indication associations

We used the well-established Relative reporting ratio (RRR) together with a $\chi^2$ statistic for disproportionality signal detection (*Harpaz et al., 2013a*). We constructed ingredient-ADR contingency tables and calculated the expected number of occurrences, the RRR, and Yates-corrected $\chi^2$ p-values (*Yates, 1934*) for these contingency tables, as implemented in SciPy (*Jones et al., 2001*). False discovery rate (FDR) was controlled (*Jones et al., 2001*) using the Holm procedure (*Holm, 1979*), yielding q-values. Associations were selected if they: (a) were reported at least five times in FAERS; (b) had a q-value <0.05; and (c) had an RRR >1. These ingredient - ADR pairs are shown in *Supplementary file 1*.

## Calculating ingredient – ADR associations on monthly basis

With FAERS data annotated with dates of ADRs, for every ingredient – ADR pair we calculated co-occurrence frequencies, RRR-values, and $\chi^2$-based p-values for every month between January 1997 and December 2015. In these calculations, we used the numbers of drugs, ADRs, and total reports from the relevant month only. False discovery rate (FDR) was controlled using the Holm procedure (for each month separately), yielding q-values. In *Supplementary file 1* for every statistically significant (in aggregate) ingredient – ADR association, we reported the numbers of months where q-values were lower than 0.05 and where q-values were higher than or equal to 0.05.

## General aspects of statistics used for post-marketing pharmacovigilance

Since spontaneous reports are not rigorously sampled from a population of patients with known exposure to the drugs of interest, incidence rates or relative risks cannot be computed. Instead, ratios of drug-event counts to intuitively plausible exposure measures, called disproportionality ratios, are commonly computed. Databases of prescription counts seem natural to use, as discussed in literature (*Strom et al., 2013*); however, these lead to difficulties in integrating two separate databases as well as ambiguities in the meaning of prescriptions, such as measuring number of months' supply, geographic region of manufacture and dispensing, patient consumption, and so forth. Both FDA (*Duggirala et al., 2016*) and EMA (*Eudravigilance Expert Working Group, 2006*) guidance documents for analysis of spontaneous report databases by two of the world's most prominent pharmaceutical regulatory bodies, focus on the use of disproportionality measures that can be computed from within a single database of spontaneous reports. PRR (*Duggirala et al., 2016*) and the closely related RRR defined above are straightforward measures computable from the simple $2 \times 2$ table of report counts classified by drug mentioned and event mentioned, while a slightly more complex (*Mantel and Haenszel, 1959*) computation can first stratify by possibly confounding variables such as patient gender and age, and calendar year of report, and then sum across strata to get an expected count to be compared to the observed count. The FDA report (*Duggirala et al., 2016*) also discusses the concept of Bayesian 'shrinkage' estimates (*DuMouchel, 1999*), which attempt to reduce the effects of variance in small samples by fitting the entire array of drug-event disproportionalities to a prior distribution and then using that to compute a posterior distribution for each individual disproportionality, resulting in a central estimate denoted EBGM (empirical Bayes geometric mean) as well as a 90% posterior range (EB05, EB95). In this paper, we use the simple measure RRR for several global computations across the FAERS database, and in some places present the presumably more reliable Bayesian measures when focusing on particular drug-events of interest.

## Clustering of ADR by time evolution

We considered the numbers of reports in each month (time evolutions) for individual ADRs observed across FAERS for four drugs: rofecoxib, celecoxib, rosiglitazone, and pioglitazone. With knowledge of numbers of each of the considered drugs, ADRs, and the number of total reports in FAERS in each month, we calculated month-resolved RRR-values for the drug-ADR pairs. The time evolutions of the RRR-values were clustered for each drug using the partitioning method for maximum dissimilarity, as implemented in R (*R Core Team, 2013*) scored by the similarity (Pearson correlation coefficient) of time evolutions of RRR.

## Logistic regression models of myocardial infarction dependence on the use of celecoxib

For every FAERS report, we noted whether a) celecoxib was reported as the primary suspect drug, b) whether myocardial infarction was reported, c) the occupation of the person filing the report, and d) whether the reported event took place before 2005 (when rofecoxib was still on the market). Using this data and R's implementation of binomial logistic regression (via the glm() function), we prepared four models (with the logit link function) (*R Core Team, 2013*) to investigate if myocardial infarction is associated with the use of celecoxib, the occupation of the person filing the report, and the report being filed before 2005. In each model, reporting myocardial infarction served as the output variable, and combinations of the remaining variables were used as input variables. Resulting models are summarized and described in more detail in *Supplementary file 2*.

## Analysis of association between VEGF-R2 inhibition and hypertension

Apart from analysis described in this work, additional Bayesian data mining and statistical analysis of VEGF-R2 inhibition-related hypertension was based on the methods described in detail by DuMouchel (*DuMouchel, 1999*), DuMouchel and Pregibon (*DuMouchel and Pregibon, 2001*), Szarfman et al. (*Szarfman et al., 2002*), and Almenoff et al. (*Almenoff et al., 2007*).

## Acknowledgements

MM was supported in the initial stage of this work by the NIBR Presidential Postdoctoral Fellowship co-mentored by LU and BKS. The work of BKS was supported by US National Institutes of Health Grant R01 GM71896 and by R35 GM122481. We thank Dr Duncan Armstrong for his comments and advice on the manuscript.

## Additional information

### Competing interests

MM: Mateusz Maciejewski is an employee of Pfizer Inc. EL: Eugen Lounkine is an employee of Novartis Institutes for BioMedical Research. SW: Steven Whitebread is an employee of Novartis Institutes for BioMedical Research. PF: Pierre Farmer is an employee of Novartis Institutes for BioMedical Research. WD: Bill DuMouchel is an employee of Oracle Health Sciences. BKS: Brian K. Shoichet has previously consulted for Novartis. LU: Laszlo Urban is an employee of Novartis Institutes for BioMedical Research.

### Funding

| Funder | Author |
| --- | --- |
| Novartis | Mateusz Maciejewski<br>Eugen Lounkine<br>Steven Whitebread<br>Pierre Farmer<br>William DuMouchel<br>Laszlo Urban |
| National Institutes of Health | Brian K. Shoichet |

The funders had no role in study design, data collection and interpretation, or the decision to submit the work for publication.

## Author contributions

MM, Conceptualization, Data curation, Software, Formal analysis, Validation, Visualization, Methodology, Writing—original draft, Writing—review and editing; EL, Conceptualization, Software, Supervision, Validation, Writing—original draft, Writing—review and editing; SW, Validation, Methodology, Writing—original draft, Writing—review and editing; PF, Validation, Writing—review and editing; WD, Software, Validation, Methodology; BKS, Supervision, Validation, Methodology, Writing—original draft, Writing—review and editing; LU, Conceptualization, Formal analysis, Supervision, Validation, Methodology, Writing—original draft, Writing—review and editing

## Author ORCIDs

Mateusz Maciejewski, http://orcid.org/0000-0003-1147-4941
Eugen Lounkine, http://orcid.org/0000-0002-6487-2946
Laszlo Urban, http://orcid.org/0000-0002-1275-0293

## Additional files

### Supplementary files

• Supplementary file 1. A document containing multiple sheets with reported data: groups of case_ids of reports that were found to likely correspond to the same event; monthly fractions of reported ADRs in rofecoxib reports; monthly fractions of reported ADRs in celecoxib reports; monthly fractions of reported ADRs in rosiglitazone reports; monthly fractions of reported ADRs in pioglitazone reports; ingredient – ADR associations stemming from analysis of all reports; aggregation of ingredient – ADR associations resolved by month.

• Supplementary file 2. Summary of logistic regression models inspecting the dependence of reporting myocardial infarction on the use of celecoxib (Model 1: the following formula was used in R's glm() function: has_adr~has_celecoxib), occupation of the party reporting celecoxib to cause myocardial infarction (Model 2: has_adr~has_celecoxib*occupation–occupation), time a celecoxib – myocardial infarction report was filed (Model 3: has_adr~has_celecoxib*before_2005–before_2005), and a combination of occupation and time (Model 4:has_adr~has_celecoxib*occupation*before_2005–occupation*before_2005). 'Physician' was used as the reference level in Models 2 and 4 for the variables involving the occupation of the reporting party. Significance codes: 0 '***' 0.001 '**' 0.01 '*' 0.05 '.' 0.1 ''1. The ':' symbol separating two variable names denotes a variable representing an interaction between these two variables.

• Supplementary file 3. The Bayesian calculations used the program Empirica Signal and the version of FAERS offered commercially by Oracle Health Sciences. The Oracle algorithms for detection of duplicate reports and determination of generic drug names yield slightly different counts than does the data preparation method described in the present paper, although computed adverse event rates are virtually identical. The columns for N, E, and EB05 shown in the table are as calculated by the Oracle program. Columns represent: Drug (Generic name) PT (MedDRA Preferred Term) N (Number of reports including both Drug and PT, 1997–2015, in the Oracle Health Sciences curation of FAERS) E (Expected value of N if Drug and PT are independent within each stratum, where strata are defined by all combinations of gender, age group and year of report) RR (Relative Reporting Rate = N/E) EBGM (Empirical Bayes Geometric Mean of estimated disproportionality) EB05 (Lower limit of Bayesian 90% confidence interval of true disproportionality) EB95 (Upper limit of Bayesian 90% confidence interval of true disproportionality)

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
