## [Decision Letter]

Thank you for submitting your article "Reverse Translation of Adverse Event Reports Paves the Way for De-risking Preclinical Off-Targets" for consideration by eLife. Your article has been reviewed by two peer reviewers, and the evaluation has been overseen by Fiona Watt as the Reviewing and Senior Editor. The following individuals involved in review of your submission have agreed to reveal their identity: Ellen Berg (Reviewer #1); Keith Houck (Reviewer #2).

The reviewers have discussed the reviews with one another and the Reviewing Editor has drafted this decision to help you prepare a revised submission.

Summary:

This manuscript is a welcome in-depth study that investigates reports from the FDA's Adverse Event Reporting System (FAERS). In this era of big data, a large database such as FAERS, developed for post-marketing pharmacovigilance and containing more than 8.5 million reports, is enticing to mine for potential connections between drugs or drug classes and specific adverse events.

The authors have a successful track record of leveraging large databases of secondary pharmacology assay data to identify novel target-ADR connections, so this is an obvious extension of this work. There are critical differences, however. Secondary pharmacology assays are run under standardized conditions typically under the control of sponsor organizations. Data completeness and quality are substantially higher.

The authors recognize these differences and provide a quite thoughtful and useful investigation of issues related to the use of FAERS data. The issues cover: (1) [13]drug naming and synonym aggregation; (2) [85]errors in data entry; (3) [39]diversity and patterns in the data depending on the report submitters (patients and lawyers in addition to health care professionals); and (4) [26]reporting influences, such as news events and changes in prescription guidelines over time.

Maciejewski et al. walk through these issues with detailed analysis on overall reports, as well as ADR reports for specific drugs including drug pairs and relevant comparators. Case studies include: COX-2 inhibitors, celecoxib versus rofecoxib; thiazolidinones, rosiglitazone versus pioglitazone; VEGFR2 inhibitors; and anti-psychotics. Overall the findings are well-supported and of interest to researchers in the field.

Essential revisions:

Discussion. In the Discussion, it would be helpful to include specific recommendations for improving FAERS to facilitate this type of research, such as better classifications; improving reporting automation (to reduce misclassifying ADRs versus indications); curation; mapping drugs and synonyms to ingredients, etc. A more explicit discussion of the limitations of the approach such as the lack of prescription (exposure) information is required.

Methods and materials. The authors use RRR as a metric, which reflects the relative reporting of one type of ADR among all of the ADRs reported for that drug. The authors should discuss other metrics such as normalizing reports to the number of prescriptions (if this is possible or not, advantages, disadvantages, etc.).

---

## [Author Response]

*Essential revisions:*

*Discussion. In the Discussion, it would be helpful to include specific recommendations for improving FAERS to facilitate this type of research, such as better classifications; improving reporting automation (to reduce misclassifying ADRs versus indications); curation; mapping drugs and synonyms to ingredients, etc. A more explicit discussion of the limitations of the approach such as the lack of prescription (exposure) information is required.*

The suggestions about how to improve FAERS are well-taken (“…it would be helpful to include specific recommendations for improving FAERS…*”*). To address this critique, we have added several extensive passages to the Discussion section:

“Matters can be improved when a drug is used in different indications with distinct symptoms and outcomes, enabling differentiation between disease- and drug-related outcomes. […] For now, the category “outcome” should be used cautiously for ADR analysis, especially in large-scale studies that aggregate data from several drugs.”

We also extended the Discussion section with the following additions based on the reviewers’ recommendations:

“FAERS, an already valuable asset for clinicians and pharmaceutical scientists, could be improved in several ways to improve post-marketing pharmacovigilance. […]While this development would need investment, one could imagine undertaking it as part of a public-private partnership from which all would benefit.”

Methods and materials. The authors use RRR as a metric, which reflects the relative reporting of one type of ADR among all of the ADRs reported for that drug. The authors should discuss other metrics such as normalizing reports to the number of prescriptions (if this is possible or not, advantages, disadvantages, etc.).

A paragraph (including References) has been added that reviews the various types of disproportionality measures applied to drug-event combinations within spontaneous report databases. As stated we use a simple to compute and interpret measure RRR for many of our globally computed overviews, and more advanced Bayesian measures in a table focusing on particular combinations of interest:

Since spontaneous reports are not rigorously sampled from a population of patients with known exposure to the drugs of interest, incidence rates or relative risks cannot be computed. Instead, ratios of drug-event counts to intuitively plausible exposure measures, called disproportionality ratios, are commonly computed. […] In this paper, we use the simple measure RRR for several global computations across the FAERS database, and in some places present the presumably more reliable Bayesian measures when focusing on particular drug-events of interest.